# Bench-V: A Primary Assessment for Visual Reasoning Models with Multi-modal Outputs

**Meng-Hao Guo[1], Xuanyu Chu[1], Qianrui Yang[1], Zhe-Han Mo[1], Yiqing Shen[1]**

**Pei-Lin Li[1], Xinjie Lin[1], Jinnian Zhang[2], Xin-Sheng Chen[1], Yi Zhang[1]**

**Kiyohiro Nakayama[3], Zhengyang Geng[4], Houwen Peng[2], Han Hu[2], Shi-Min Hu[1] ***

[1] Tsinghua University, [2] Tencent Hunyuan Team, [3] Stanford University, [4] Carnegie Mellon University

## Abstract

The rapid advancement of native multi-modal models and omni-models, exemplified by GPT-4o, Gemini and o3 with their capability to **process and generate** content across modalities such as text and images, marks a significant milestone in the evolution of intelligence. Systematic evaluation of their multi-modal output capabilities in visual thinking process (*a.k.a.,* multi-modal chain of thought, M-CoT) becomes critically important. However, existing benchmarks for evaluating multi-modal models primarily focus on assessing multi-modal inputs and text-only reasoning process while neglecting the importance of reasoning through multi-modal outputs. In this paper, we present a benchmark, dubbed as $\mathcal{R}$Bench-V, designed to assess models' vision-indispensable reasoning. To conduct $\mathcal{R}$Bench-V, we carefully hand-pick 803 questions covering math, physics, counting and games. Unlike problems in previous benchmarks, which typically specify certain input modalities, $\mathcal{R}$Bench-V presents problems centered on multi-modal outputs, which require image manipulation, such as generating novel images and constructing auxiliary lines to support reasoning process. We evaluate numerous open- and closed-source models on $\mathcal{R}$Bench-V, including o3, Gemini 2.5 pro, Qwen2.5-VL, etc. Even the best-performing model, o3, achieves only 25.8% accuracy on $\mathcal{R}$Bench-V, far below the human score of 82.3%, which shows current models struggle to leverage multi-modal reasoning. Data and code are available at `https://evalmodels.github.io/rbenchv`.

## 1 Introduction

*"What I cannot create, I do not understand."* — Richard Feynman.

Whether adults or children, when faced with complex problems, they sometimes turn to drawing or diagramming to organize their thoughts, support reasoning, and seek solutions. As highlighted by the quote 1 and findings in neuroscience Goldschmidt [1991], Pylyshyn [2001], Edwards [2012], Fan et al. [2023], the capability to draw during problem-solving is not only a hallmark of cognitive development but also an expression of human intelligence. But what about intelligent models? Can they also learn to draw in order to reason and solve problems?

Recently, researchers have made great progress toward equipping foundation models with above capability, and the landscape of foundation models has undergone a profound transformation, driven

---

*Shi-Min Hu is the corresponding author. E-mail: shimin@tsinghua.edu.cn.

39th Conference on Neural Information Processing Systems (NeurIPS 2025) Track on Datasets and Benchmarks.

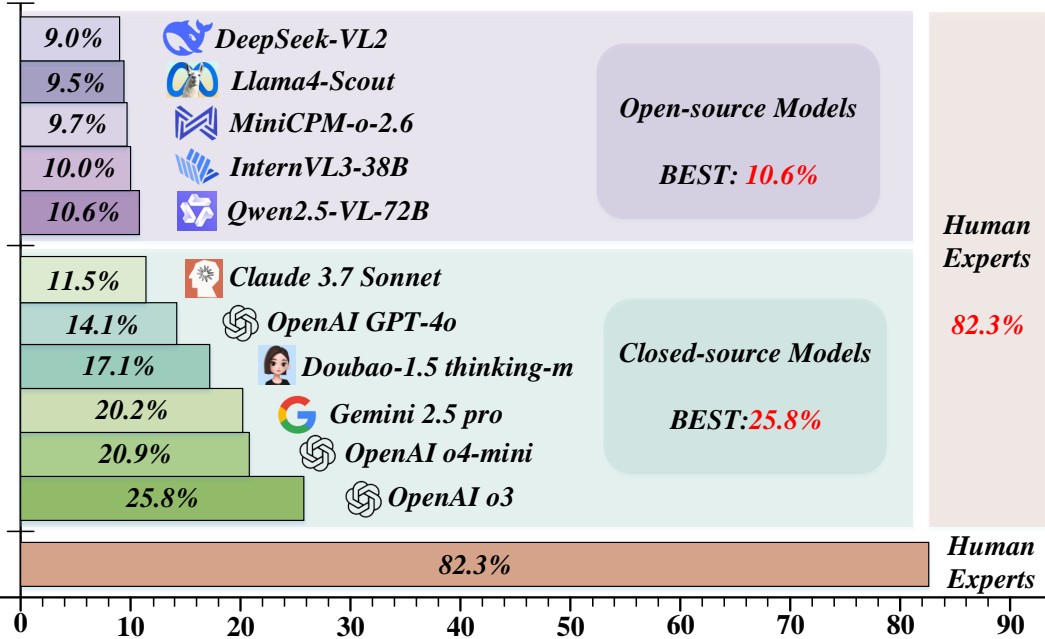

Figure 1: The comparison between open-source models, closed-source models and human experts on $\mathcal{R}$Bench-V. It reveals there remains a significant gap between models and human experts in visual reasoning with multi-modal outputs.

by two major converging trends. First, *modal convergence* reflects the evolution from single-modality language models, such as ChatGPT OpenAI [2022], to omni-modal systems capable of both multi-modal input and output, exemplified by GPT-4o OpenAI [2024a]. Second, *cognitive convergence* captures the transition from chat-oriented models to reasoning-driven models, as evidenced by the progression from ChatGPT OpenAI [2022] to more advanced systems such as OpenAI o1/o3 OpenAI [2024b] and DeepSeek R1 DeepSeek [2025].

As model input and output modalities converge, the evaluation frameworks for these leading foundation models must also evolve accordingly. Existing benchmarks such as MMMU Yue et al. [2024a] and MMLU Hendrycks et al. [2021], have played a key role in advancing the field by providing frameworks to evaluate model capabilities. However, these benchmarks are predominantly input-oriented, focusing on the model's ability to interpret, understand, and reason over multi-modal inputs, while overlooking an equally critical aspect: the modality of outputs. This refers to the model's ability to generate contextually appropriate multi-modal responses during the problem-solving process, whether through language, visual content, or other formats.

In this paper, we present $\mathcal{R}$Bench-V, an early exploration for multi-modal output-oriented reasoning benchmark. To build $\mathcal{R}$Bench-V, we carefully and strictly hand-pick and design 803 question-answer pairs, covering math, physics, counting, and games. In Fig 3, we clearly illustrate the differences between $\mathcal{R}$Bench-V and other classic language model benchmarks, MMLU Hendrycks et al. [2021], and the multi-modal input-oriented benchmark, MMMU Yue et al. [2024a]. It can be seen that the main difference from previous benchmarks is that in $\mathcal{R}$Bench-V, each question requires the model to produce multi-modal outputs during the reasoning process, particularly modifications on the images, such as drawing images, adding auxiliary lines, and so on.

We evaluate a wide range of open- and closed-source multi-modal large language models (MLLMs) and omni models on the $\mathcal{R}$Bench-V, including GPT-4o OpenAI [2024a], Gemini Google et al. [2023], Qwen2.5VL Bai et al. [2025], Claude3.5 Anthropic [2024], DeepSeek-VL2 Wu et al. [2024], *etc.* Besides, we also organize human to conduct tests on $\mathcal{R}$Bench-V. Our main observations and findings from the experiments are highlighted as follows. For more details, please refer to the experiment section.

- If models, such as the InternVL or Qwen-VL series, lack multi-modal CoT, merely increasing their model sizes will not effectively resolve the challenges of vision-indisperential reasoning.

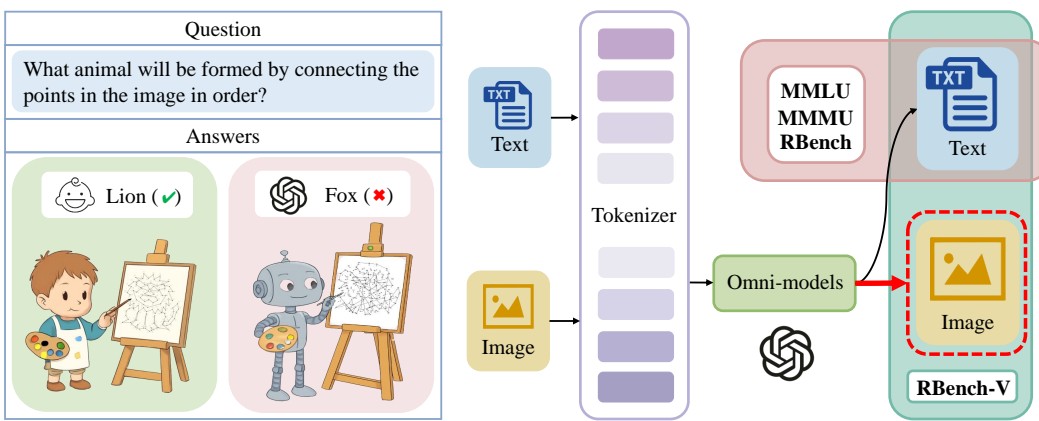

Figure 2: The motivation of $\mathcal{R}$Bench-V. Left: An illustration showing both humans and the GPT-4o model being asked a game-related question from $\mathcal{R}$Bench-V. Right: This part shows common benchmarks such as MMLU, MMMU, and Rench focus on multi-modal inputs and textual outputs, whereas $\mathcal{R}$Bench-V emphasizes not only multi-modal inputs but also multi-modal outputs.

It might be necessary to explore new paradigms, potentially incorporating M-CoT or agent-based reasoning frameworks, to solve visual-necessary complex problems.

- Despite the diverse capabilities of these models, even the highest-performing one, *i.e.*, o3 OpenAI [2025c], achieves only 25.8% accuracy on $\mathcal{R}$Bench-V, which is significantly lower than the human score of 82.3%. This stark contrast highlights that, while impressive in many areas, the models still struggle to generate and integrate appropriate multi-modal responses in the visual thinking process. Besides, the o3 model has achieved significant progress in visual reasoning, outperforming previous state-of-the-art models by a substantial margin (+4.9%). $\mathcal{R}$Bench-V effectively captures this advancement and offers an automated framework for evaluating multi-modal output capabilities in visual reasoning.

- The reasoning thought of models differ from that of human experts in math. We find that while models perform well on math questions, this does not necessarily indicate that they have learned to multi-modal reasoning. Instead, models often convert certain geometry problems into algebraic ones and solving them through textual reasoning. In contrast, human experts tend to prefer geometric solutions. This highlights a fundamental difference between the intelligence exhibited by current models and that of humans.

## 2 Related Work

### 2.1 Foundation Models

Foundation models have evolved rapidly along two axes: the expansion from language understanding to omni-models, and the progression from chat models to advanced reasoning models.

**From language models to omni-models:** It represents progress along the modality axis. Early foundation models—such as ChatGPT OpenAI [2022], LLaMA Touvron et al. [2023], Qwen Bai et al. [2023], Mistral Jiang et al. [2023], GLM Zeng et al. [2022], *etc*, are limited to text-based dialogue. Researchers then begin exploring models with multi-modal inputs, including GPT-4V OpenAI [2023], LLaVA Liu et al. [2024], miniGPT-4 Zhu et al. [2023], Claude3.5 Anthropic [2024], *etc*. Recently, attention has shifted toward omni-models, which not only receive multi-modal inputs but also generate flexible multi-modal outputs (e.g., GPT-4o OpenAI [2024a], Gemini 2.5 Pro Google [2025], and Emu3 Wang et al. [2024b]).

**From chat models to reasoning models:** Another critical axis of advancement lies in reasoning capabilities. Early chat-based foundation models OpenAI [2022], Touvron et al. [2023], Abdin et al. [2024] primarily focus on fluent and context-aware dialogue generation. Recently, researchers have begun to push the boundaries of model reasoning OpenAI [2024b], DeepSeek [2025], Google [2025],

aiming to equip models with the ability to synthesize existing knowledge and solve complex, novel problems. For more, readers are referred to this survey Wang et al. [2025].

## 2.2 Benchmarking Foundation Models

Benchmarks serve as an essential tool for evaluating foundation models and providing a guiding light for their further development. Existing benchmarks primarily focus on multi-modal inputs such as text Hendrycks et al. [2021], Guo et al. [2025], Wang et al. [2024c], Chen et al. [2021], Contributors [2023] and multimodality Yue et al. [2024a,b], Lu et al. [2023], Wang et al. [2024a], Lu et al. [2023], Gao et al. [2024], but their outputs are all textual, overlooking the evaluation of models' multi-modal output capabilities. Besides, some studies Heusel et al. [2017], Kastryulin et al. [2022], You et al. [2024] have also focused on image generation quality, but they primarily emphasize aesthetic metrics.

As mentioned in Sec. 2.1, we believe that the next generation of powerful models are omni-models with strong textual and visual reasoning capabilities. Thus, in this paper, we present $\mathcal{R}$Bench-V, a benchmark for omni reasoning models. To the best of our knowledge, this is the first attempt to design benchmark to assess models' multi-modal generation capability in the visual thinking process.

# 3 $\mathcal{R}$Bench-V

## 3.1 Data Collection of $\mathcal{R}$Bench-V

The central challenge in developing $\mathcal{R}$Bench-V lies in designing and curating questions that assess models' ability to generate multi-modal outputs during visual reasoning. Clear examples are shown in Fig. 3, solving problems in $\mathcal{R}$Bench-V requires producing outputs beyond text, such as drawing geometric figures (top-right example) or tracing path through a maze (bottom-right example).

To build $\mathcal{R}$Bench-V, our principle in designing or collecting questions is that their solution should involve creating new visual content, such as creating images or modification of existing images, during the problem-solving process. We can imagine that numerous real-world scenarios, such as GUI operation and drawing, rely on multi-modal outputs for visual reasoning in daily life. In this work, $\mathcal{R}$Bench-V primarily focuses on math, physics, counting, and games. To curate high-quality questions in math and physics, we collaborate with domain experts. For counting and games, we conduct a rigorous rule to create, collect, and filter questions. The collection criteria for different domains are detailed as follows.

- **Math:** For math, we primarily focus on geometric and graph theory problems, including transformation geometry, planar geometry, solid geometry, *etc*. **Transformation Geometry:** The problems in $\mathcal{R}$Bench-V mainly involve translations, reflections and rotations, requiring the model to draw the resulting figures after applying these transformations. in order to arrive at the correct answer. **Planar Geometry:** These problems evaluate whether the model can construct appropriate auxiliary lines to aid in reasoning. **Solid Geometry:** The solid geometry tasks in $\mathcal{R}$Bench-V assess models to assemble 3D shapes from 2D components according to specific rules and to draw the resulting solid before answering the question. **Graph Theory:** $\mathcal{R}$Bench-V requires models to first complete the graph based on given constraints, mark the leaf nodes, and then reason to arrive at the correct answer.

- **Physics:** We primarily focus on optics, mechanics, electromagnetism, and thermodynamics. Not all above problems meet our criteria and we specifically select those that require visual reasoning. **Optics:** Tasks emphasize geometric optics, requiring models to trace light trajectories involving reflection, refraction, and diffraction. Precise visualization of light paths is essential for deriving optical principles. **Mechanics:** This category includes statics, kinematics, and dynamics, involving complex physical constraints. Models must interpret and construct geometric relationships using free-body diagrams and motion trajectories to analyze force interactions, motion paths, and equilibrium conditions. **Electromagnetism:** This area comprises two subcategories. Circuit analysis tasks require models to identify current paths and simplify circuit diagrams in complex scenarios. Dynamic problems combine electromagnetic fields with mechanics, necessitating the visualization of electric and magnetic field lines to analyze particle motion. **Thermodynamics:** Tasks primarily involve fluid force analysis, where models must visually represent dynamic changes in liquid surfaces and force distributions to solve problems related to surface tension, hydrostatic pressure, and buoyancy.

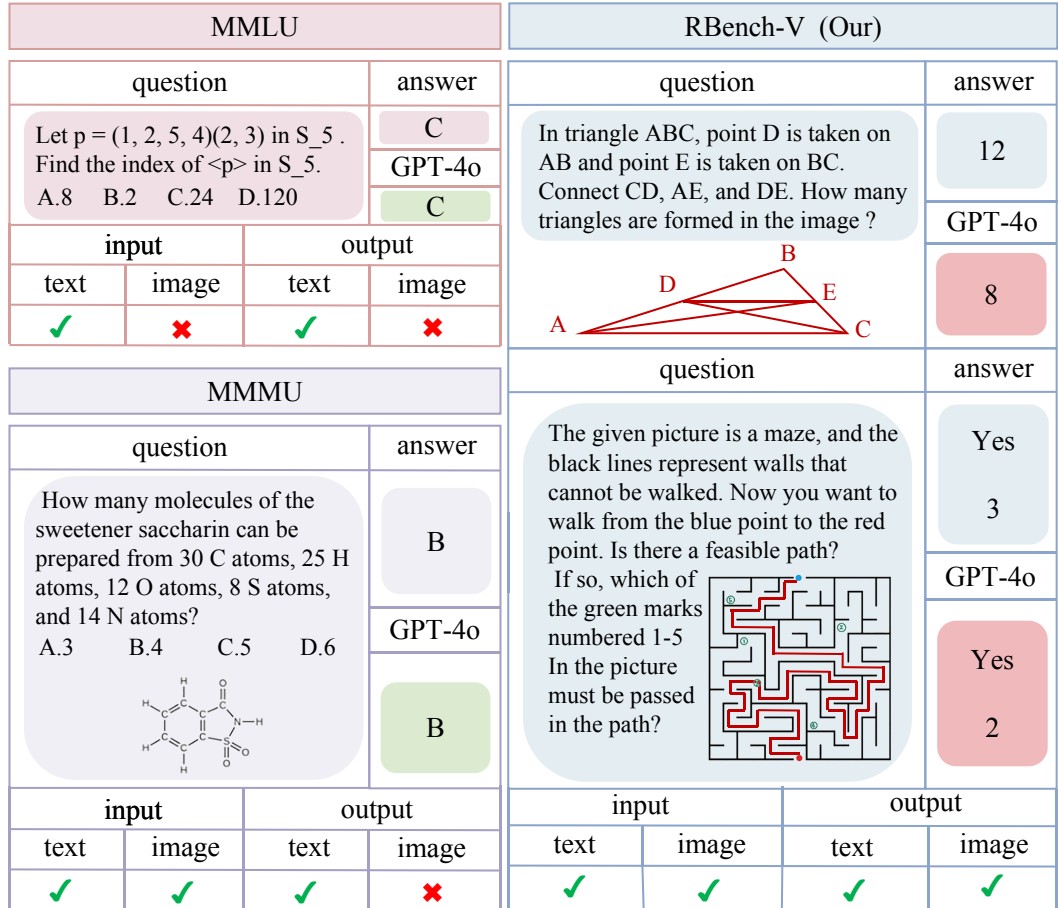

Figure 3: A visual comparison with MMLU, MMMU and $\mathcal{R}$Bench-V. It shows that solving problems in MMLU and MMMU mainly requires understanding multi-modal inputs and generating textual outputs, whereas solving problems in $\mathcal{R}$Bench-V demands not only understanding multi-modal inputs but also generating multi-modal outputs. The **red** lines shown in the figure are **not** part of the original questions and represent the multi-modal reasoning process when solving problems in $\mathcal{R}$Bench-V, such as drawing geometric shapes or tracing paths through a maze.

- **Counting:** Counting problems typically require no image modification. For instance, identifying two people in a clearly depicted photo needs no interaction. The counting problems in $\mathcal{R}$Bench-V differ from the simple example above and can be broadly categorized into the following three types: **Firstly,** problems require drawing geometric shapes based on descriptions or connecting lines within the image before answering questions such as how many triangles are present (an example is shown in the top-right corner of Fig. 3). **Secondly,** questions involve images with lots of targets and chaotic backgrounds. Solving such problems requires models to carefully check the images, mark the targets it has counted, and then reason to answer the total. (examples are provided in the supplementary materials.) **Thirdly,** problems demand an understanding of spatial relationships and imagination. Models need to mentally manipulate or move 3D objects and visualize the resulting state after movement, in order to complete the counting task.

- **Games:** We primarily focus on several types of games that require multi-modal outputs in the visual reasoning process: **Connect-the-dots**: Models need to connect a sequence of dots to reveal an image and identify what object in the image. **Mazes:** Models should trace the correct path through the maze and answer questions based on the trajectory. **Dart-and-balloon, and Gold Miner:** These require models to accurately draw the trajectory of darts and hooks, and determine their intersection with the target objects. **Puzzles:** The task involves moving different pieces to complete the full puzzle. **Ball-and-brick:** It requires drawing the trajectory of the ball, which may collide and bounce against the wall multiple times.

Table 2: Models and experts scores of multi-modal output requirements across different benchmarks.

(a) $\mathcal{R}$Bench-V vs. MMLU on text-only questions.

|  | $\mathcal{R}$Bench-V win | MMLU win | Tie |
|---|---|---|---|
| Model | 92.4 | 3.8 | 3.8 |
| Human | 86.4 | 4.5 | 9.1 |

(b) $\mathcal{R}$Bench-V vs. MMMU on multi-modal questions.

|  | $\mathcal{R}$Bench-V win | MMMU win | Tie |
|---|---|---|---|
| Model | 94.1 | 5.9 | 0.0 |
| Human | 83.4 | 6.6 | 10.0 |

## 3.2 Statistics of $\mathcal{R}$Bench-V

We conduct statistical analysis on $\mathcal{R}$Bench-V with the results presented in Tab. 1. It presents that $\mathcal{R}$Bench-V includes 803 questions across four areas, with 176 math questions, 157 physics questions, 195 counting problems, and 275 game-related questions, comprising 356 multiple-choice questions (We categorize questions with clearly limited answer choices as multiple-choice questions, such as the maze problem in the bottom right of Fig. 3.) and 447 open-ended questions. It is worth noting that since we primarily focus on multi-modal outputs rather than inputs, so $\mathcal{R}$Bench-V includes both text-only and multi-modal input questions. categorized as 40 and 763, respectively. As an early exploration into visual reasoning and multi-modal outputs, this paper focuses on text and image modalities, aiming to offer insights for foundation models. As for more modalities such as video and audio outputs, we expect more work to be done in the future.

Table 1: Statistics on $\mathcal{R}$Bench-V. MC denotes multiple-choice.

| Statistics | Number |
|---|---|
| Total Questions | 803 |
| Math Questions | 176 |
| Physics Questions | 157 |
| Counting Questions | 195 |
| Games Questions | 275 |
| MC Questions | 356 |
| Open Questions | 447 |
| Text-only Questions | 40 |
| Multi-modal Questions | 763 |

## 4 Experiments

After developing $\mathcal{R}$Bench-V, we comprehensively evaluate many open- and closed-source MLLMs (*e.g.,* Qwen2.5VL Bai et al. [2025], Claude3.7 Anthropic [2024], LLaVA-OneVision Li et al. [2024], *etc*) , omni-models (*e.g.,* GPT-4o OpenAI [2024a], Gemini2.5pro Google et al. [2024], Qwen2.5-Omni Xu et al. [2025], *etc*) , thinking models (*e.g.,* OpenAI o3 OpenAI [2025c] and Doubao-1.5-thinking-pro-m Seed [2025b]) and humam experts on $\mathcal{R}$Bench-V. We list all the tested models in Tab. 3.

By default, all evaluations are conducted in a zero-shot setting. Besides, since $\mathcal{R}$Bench-V includes both multiple-choice and open-ended questions, we adopt a unified LLM-as-a-Judge framework, with the judge model being GPT-4o. We report Top-1 accuracy (%) as our default evaluation metric.

### 4.1 Comparison with other benchmarks on multi-modal output capability requirements

Here, we compare $\mathcal{R}$Bench-V with other benchmarks (MMLU, MMMU) in terms of multi-modal output evaluation. As we know, it is challenging to find a quantitative method to assess the property for multi-modal outputs in existing benchmarks. Therefore, we construct expert scores and model scores to measure their multi-modal reasoning property. Specifically, we randomly sample 30 examples from each of $\mathcal{R}$Bench-V, MMLU, and MMMU, and instruct either human experts or the models (o3 and Doubao-1.5-thinking-m) to compare whether requires drawing during the thinking process.

We summarize the win rates in Table 2. Results from both human experts and models consistently show that $\mathcal{R}$Bench-V imposes a significantly higher requirement for multi-modal outputs during the reasoning process compared to MMLU and MMMU. This highlights that $\mathcal{R}$Bench-V is specifically designed to assess multi-modal output capabilities and visual reasoning skills.

### 4.2 Evaluating visual reasoning and multi-modal outputs of different models

We assess various open- and closed-source models, along with human experts, on $\mathcal{R}$Bench-V. The specific models are listed in Tab. 3. For open-source models, we use vLLM Kwon et al. [2023] and VLMEvalKit Duan et al. [2024] for deployment, setting the temperature as 0 while leaving all other

Table 3: Performance (%) of different models and human experts on $\mathcal{R}$Bench-V. † means long chain thinking model. The highest scores are highlighted in **red**, and the second-highest scores are highlighted in **blue**.

| Name | Overall | w/o Math | Math | Physics | Counting | Games |
|---|---|---|---|---|---|---|
| Open-source models | | | | | | |
| Qwen2.5-Omni-7B  Xu et al. [2025] | 7.7 | 4.5 | 11.4 | 1.9 | 2.1 | 7.7 |
| InternVL-3-14B Zhu et al. [2025] | 8.0 | 7.0 | 11.4 | 1.3 | 5.1 | 11.6 |
| InternVL-3-8B Zhu et al. [2025] | 8.2 | 6.0 | 15.9 | 1.9 | 5.6 | 8.7 |
| Qwen2.5VL-7B Bai et al. [2025] | 8.3 | 7.0 | 13.1 | 2.5 | 3.6 | 12.0 |
| LLaVA-OneVision-7B Li et al. [2024] | 8.5 | 6.8 | 14.2 | 2.5 | 4.6 | 10.9 |
| DeepSeek-VL2 Wu et al. [2024] | 9.0 | 7.0 | 15.9 | 0.6 | 5.6 | 11.6 |
| LLaVA-OneVision-72B Li et al. [2024] | 9.0 | 8.9 | 9.1 | 4.5 | 4.6 | 14.5 |
| MiniCPM-2.6-V Yao et al. [2024] | 9.1 | 7.2 | 15.9 | 1.3 | 6.2 | 11.3 |
| Llama4-Scout (109B MoE) Meta [2025] | 9.5 | 6.9 | 18.8 | 3.2 | 4.1 | 10.9 |
| MiniCPM-2.6-o Yao et al. [2024] | 9.7 | 7.5 | 17.6 | 1.3 | 3.6 | 13.8 |
| Qwen2.5VL-32B Bai et al. [2025] | 10.0 | 6.4 | 22.7 | 2.5 | 4.1 | 10.2 |
| InternVL-3-38B Zhu et al. [2025] | 10.0 | 7.2 | 20.5 | 0.6 | 5.1 | 12.4 |
| Qwen2.5VL-72B Bai et al. [2025] | 10.6 | 9.2 | 15.3 | 3.8 | 6.2 | 14.5 |
| Closed-source models | | | | | | |
| QVQ-Max Qwen [2025] | 11.0 | 8.1 | 21.0 | 5.7 | 6.2 | 10.9 |
| Claude-3.7-sonnet Anthropic [2025] | 11.5 | 9.1 | 19.9 | 3.8 | 8.7 | 12.4 |
| OpenAI GPT-4.5 OpenAI [2025b] | 12.6 | 11.0 | 18.2 | 2.5 | 11.8 | 15.3 |
| Step-R1-V-Mini† StepFun [2025] | 13.2 | 8.8 | 29.0 | 6.4 | 10.3 | 9.1 |
| OpenAI GPT-4.1 OpenAI [2025a] | 13.6 | 11.7 | 20.5 | 5.7 | 11.3 | 15.3 |
| OpenAI GPT-4o-20250327 OpenAI [2024a] | 14.1 | 11.2 | 24.4 | 3.2 | 13.3 | 14.2 |
| Doubao-1.5-vision-pro Seed [2025a] | 15.6 | 11.5 | 30.1 | 8.9 | 12.8 | 12.0 |
| OpenAI o1† OpenAI [2024b] | 16.2 | 11.0 | 34.7 | 5.7 | 12.3 | 13.1 |
| Doubao-1.5-thinking-pro-m† Seed [2025b] | 17.1 | 11.0 | 38.6 | 13.4 | 9.7 | 10.5 |
| Gemini 2.5 pro-preview-0506  Google [2025] | 20.2 | 13.9 | 42.6 | 9.6 | 19.0 | 12.7 |
| OpenAI o4-mini† OpenAI [2025c] | 20.9 | 14.6 | 43.2 | 12.7 | 17.4 | 13.8 |
| OpenAI o3† OpenAI [2025c] | **25.8** | **19.5** | **48.3** | **20.4** | **22.1** | **17.1** |
| Human Experts | | | | | | |
| Human Experts Score | **82.3** | **81.7** | **84.7** | **69.4** | **81.0** | **89.1** |

parameters at their default values. The above experiments are conducted on 8 x NVIDIA H20 GPUs. For closed-source models, we follow the official API usage guidelines provided for each model. If the official API allows setting the temperature parameter, we set it to 0; all other parameters are kept as recommended by the official documentation. For the human expert score, we invite senior undergraduate students to serve as human experts. For physics and math, we recruit some senior undergraduates for each major, assigning them different sets of questions. For games and counting tasks, we similarly invite some senior undergraduates, without restricting their academic backgrounds. The experimental results are summarized in Tab. 3, and a comprehensive analysis is presented in Sec. 4.4.

## 4.3  Visualization

Here, we visualize a correct example and an incorrect example: the correct case is from planar geometry in math, while the incorrect case is from games. The answer comes from the representative model, o3, the results are shown in Fig. 4.

From the results, we observe that although o3 arrives at the correct answer for the planar geometry question, its solution is based on an algebraic approach by establishing a coordinate system, rather than using a typical geometric method commonly adopted by humans. This suggests that the model tends to favor algebraic solutions with text-only reasoning thought over multi-modal geometric reasoning path when both approaches are available. It indicates that improvements in mathematical

<table>
<tr><td>

**Question:**
Quadrilateral ABCD is a rectangle and quadrilateral AEFG is a square with a side length of 10; given that the area of the shaded region is 18, find the area of rectangle ABCD.

**Ground Truth: 64**

</td><td>

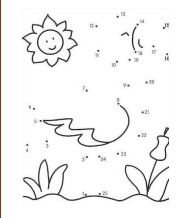

</td></tr>
</table>

**Question:**
Connect the black dots in the image sequentially with straight lines according to the edge numbers (i.e., connect dot 1 to dot 2, dot 2 to dot 3, and so on). The final result will form a simple line drawing. What does this drawing represent?

A: Duck B: Fish C: Dog D: Horse E: Cow F: Cat

**Ground Truth: A**

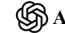

**Answer of o3:**

**Step 1 -** Let $A(0, h), B(0, 0), C(w, 0), D(w, h)$ (rectangle ABCD), $E(e, 0)$ (E is on BC). The square AEFG has side $AE = 10$, so
$$e^2 + h^2 = 10^2 = 100 \quad (1)$$
**Step 2 -** To obtain the other vertices of the square, rotate vector $AE = \langle e, -h \rangle$ 90° clockwise:
$$EF = \langle h, e \rangle, FG = \langle -h, -e \rangle, GA = \langle -e, h \rangle$$
**Step 3 -** Hence $F(e + h, e), G(h, h + e)$. The shaded region is triangle DFG. Using the determinant (cross-product) formula, twice its area is
$$|DF \times DG| = |(F - D) \times (G - D)|$$
$$= |\langle e + h - w, e - h \rangle \times \langle h - w, e \rangle|$$
$$= |e^2 + h^2 - hw|$$
**Step 4 -** Given that this area is $18, \frac{1}{2}|e^2 + h^2 - hw| = 18 \Rightarrow$
$$|e^2 + h^2 - hw| = 36 \quad (2)$$
**Step 5 -** Substituting (1) into (2):
$|100 - hw| = 36 \Rightarrow hw = 64$ or $hw = 136$. Geometry of the figure demands F be to the right of D, i.e. $e + h > w$.
If $hw = 136$, then $w = \frac{136}{h}$, and for every positive h we get $e + h < w$, placing F to the left of D, contrary to the diagram.
**Thus the compatible solution is Area of rectangle ABCD = 64.** ✔

**Answer of o3:**

**Step 1 - Observe the background drawing**
- The scene already contains water-line waves, aquatic plants and a cattail.
- Such elements suggest the main figure will be something that belongs in water.

**Step 2 - Examine the partial solid lines that are already provided along the dot path**
- Around dots 4–6 a three-lobed zig-zag shape is already drawn; this strongly resembles the forked tail of a fish.
- From dot 6 the dotted path curves upward and forward as one would draw the fish's back and then finishes near dot 25 under the water-line, matching the outline of a typical fish body.

**Step 3 - Eliminate the multiple-choice options using these observations**
- A Duck – ducks have wings and a beak above water; the pre-drawn tail is wrong.
- B Fish – matches the forked tail, curved body, and aquatic setting.
- C Dog / D Horse / E Cow / F Cat – terrestrial animals; nothing in the background supports them.

**Therefore the only option consistent with the dot-to-dot outline and the setting is "Fish."** ✘

Figure 4: Examples of o3's responses to math and game questions in $\mathcal{R}$Bench-V. Left: o3 correctly answers a math question in $\mathcal{R}$Bench-V by transforming the geometry problem into an algebraic one using a coordinate system, whereas humans typically solve it using geometric methods. Right: o3 fails to answer a game question correctly. The blue highlights indicate the cause of the error and the key issue is that the model fails to follow the instructions to draw the required connections.

performance do not necessarily reflect genuine advancements in multimodal reasoning ability, but rather suggest that models have learned certain "multi-modal reasoning shortcuts". Experts in mathematics have validated this hypothesis, pointing out that most geometry problems can be solved using algebraic methods. In contrast, counting, and games do not exhibit such "multimodal reasoning shortcuts". Therefore, we also report the performance under the "w/o math" setting in Table 3, which may serve as a better indicator of a model's true multimodal reasoning capability.

In the second question, derived from a connect-the-dots task in the games category, o3 fails to generate a correct answer. Analysis reveals that the errors here mainly stem from o3 merely attempting to describe the points in the diagram, rather than actually connecting them as required by the question. Due to space limitations, we are unable to present more examples, but our analysis shows that the majority of model failures are caused by this limitation.

### 4.4 Observation and findings

*If models, such as the InternVL or Qwen-VL series, lack multi-modal CoT, merely increasing their model sizes will not effectively resolve the challenge of visual reasoning.* As shown in Tab. 3, increasing the parameter size of the Qwen2.5VL model from 7B to 72B does not result in a clear performance improvement on $\mathcal{R}$Bench-V. A similar phenomenon is also observed in the InternVL and LLaVA-OneVision series. It suggests that the scaling law may be insufficient to address the challenges of multi-modal output in visual reasoning. Furthermore, we question whether foundation models trained primarily via next-token prediction are inherently limited in their ability to handle such tasks. While this training paradigm is well-suited for text generation, it may be fundamentally inadequate for detailed and precise multi-modal generation and understanding such as precisely tracing curve trajectory in mazes.

*Long text-only CoT approaches also do not show significant improvements on this task.* As shown in Table 3, long text-only thinking models also show only marginal gains on $\mathcal{R}$Bench-V, as evidenced by the comparison between Double1.5-vision-pro and Doubao1.5-thinking-pro-m. Combining our analysis on scaling laws, omni-models, and long text-only CoT approaches, indicating that for current foundation models, novel techniques such as M-CoT and agents can be required to effectively solve visual reasoning tasks involving precise multimodal outputs.

*Foundation models still fall well short of human expert performance in generating multi-modal outputs during visual reasoning.* As shown in Table 3, even the best-performing model to date, o3, achieves only an overall accuracy of 25.8%, which remains significantly behind the human expert score of 82.3%. This substantial performance gap underscores the limitations of current foundation models in handling tasks that demand precise multi-modal outputs in the visual reasoning process. This phenomenon is clearly illustrated by the bar chart in Fig. 1. The results emphasize that, despite recent progress, there is still considerable room for advancement in multimodal reasoning.

*The methods used by human experts and models to solve problems are not consistent.* As shown in the Tab. 3, various models tend to perform best on the mathematics subject. We analyze and present representative examples from mathematics in Fig. 4, revealing that models often convert geometric problems into algebraic ones by constructing coordinate systems. This approach differs significantly from that of human experts, who typically solve such problems using geometric reasoning. It suggests that the intelligence of current models differs from that of humans. Therefore, to avoid such "multi-modal reasoning shortcuts," we also report the accuracy after removing math-related questions in Tab 3. The results show that excluding math further amplifies the performance gap between models and human experts.

*OpenAI o3 has made substantial progress in visual reasoning with multi-modal output.* The release of o3 attracted widespread attention, largely due to its impressive ability to handle complex visual reasoning tasks—a capability that has been challenging for previous models. We observed the same phenomenon in our proposed $\mathcal{R}$Bench-V, where o3 significantly outperformed all other models in tasks that require accurate and coherent multimodal outputs. This performance lead suggests that o3 has undergone deliberate and effective enhancements specifically aimed at improving its visual reasoning and output alignment capabilities. Notably, the results also validate the design of $\mathcal{R}$Bench-V itself. It demonstrates $\mathcal{R}$Bench-V can serve as a reliable benchmark for evaluating progress and tracking how models are evolving toward human-level multimodal reasoning.

*Open-source models still lag far behind closed-source models.* Although open-source models such as Qwen2.5VL Bai et al. [2025] and LLaMA 4 Meta [2025] are making continuous progress, there remains a noticeable gap (10.6% vs. 25.8%) between open-source and closed-source models in visual reasoning tasks that require multi-modal outputs. In addition, we find that the performance of current open-source models is quite similar, with low accuracy rates mostly ranging between 8% and 10%. It suggests that current open-source models exhibit only minimal capability in multi-modal reasoning. We hope the community will develop new techniques based on open-source models to enhance their multi-modal output capabilities and ultimately close the gap with closed-source models on $\mathcal{R}$Bench-V.

*Text-based shortcuts remain a pervasive confound in assessing visual reasoning.* Despite high aggregate scores, our analyses reveal that models can often bypass genuine visual reasoning by leveraging prior knowledge and algebraic reformulations, thereby inflating performance on ostensibly "visual" tasks. This also reflects a fundamental challenge in the field today: the persistent conflation of knowledge and reasoning. Many models solve problems through memorization rather than genuine reasoning. This issue is particularly evident in $\mathcal{R}$Bench-V, where most models lack the ability to draw or visualize solutions yet can still solve certain problem, strong evidence of this shortcut phenomenon. Meanwhile, this may also reflect the misalignment between the thinking patterns of foundation models and humans. foundation models tend to prefer solving problems through textual reasoning rather than visual representations, which highlights a key challenge currently faced by MLLMs.

## 5    Conclusion

In this work, we carefully hand-pick 803 questions across 4 topics and propose $\mathcal{R}$Bench-V, a benchmark specifically designed to evaluate models' multimodal output capabilities in the visual reasoning process. It systematically assesses the current performance of models, highlights the

progress made by the o3 model in this domain, and reveals the significant gap between current intelligent models and human experts. Besides, according to our observation, the current technologies such as scaling law, long text-only CoT and joint text-visual decoding, fail to effectively address the challenges posed by $\mathcal{R}$Bench-V.

Looking ahead, we plan to further advance foundation models toward omni reasoning models, enabling more comprehensive and robust reasoning capabilities across modalities and tasks, and achieving stronger performance on $\mathcal{R}$Bench-V. In parallel, we will explore M-CoT (Multi-modal Chain-of-Thought) and agent-based strategies to enhance the reasoning depth and adaptability of these models. Furthermore, $\mathcal{R}$Bench-V is limited to text and image modalities. Future work will extend it to support richer output types, including audio, video, and other modalities, enabling more realistic and challenging multi-modal reasoning scenarios.

## 6 Acknowledgements

This work was supported by Beijing National Research Center for Information Science and Technology (BNRist). the National Natural Science Foundation of China (project No. 62495060, 623B2057), the Research Grant of Tsinghua-Tencent Joint Laboratory for Internet Innovation Technology.

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
