# OpenReview forum: "RBench-V: A Primary Assessment for Visual Reasoning Models with Multimodal Outputs"
_NeurIPS.cc/2025/Datasets_and_Benchmarks_Track — NeurIPS 2025 Datasets and Benchmarks Track poster_

### Official Review · Reviewer_REov · 2025-06-23

**Rating:** 5
**Confidence:** 4

**Summary:**

This paper introduces a new benchmark aimed at evaluating the visual reasoning capabilities of large language models (LLMs) in a multimodal context. Unlike prior benchmarks that are technically multimodal but largely test perception-related skills such as image captioning, object localization, or even image understanding, this benchmark emphasizes more abstract and structured reasoning over visual input and focus on type of questions that solving them particularly requires reasoning over the image in a more active manner compared to previous benchmarks. The authors evaluate both open-source and closed-source models, including GPT-4 (o3), and conduct human evaluations to contextualize model performance. The results show that even state-of-the-art models struggle significantly on this benchmark.

**Dataset Code Accessibility:**

Yes

**Ethical Considerations:**

No, there are no or only very minor ethics concerns

**Final Justification:**

In my review, I have raised points about more details regarding the benchmark construction, limitations of the dataset provided in the paper, as well as a shortage of qualitative analysis that would provide readers with an understanding beyond the already provided quantitative analysis in the paper.

The authors provided details that I was looking for, to address all the raised points; and I mentioned in my final comment, that given more time before the camera-ready version, they could potentially even make the paper better given the raised points that addressing them would basically add more detail into the paper and enhance the usability for future works on top of this work.

Hence, I raised my score.

**Limitations Weaknesses:**

For a dataset-centric paper, there is insufficient detail about the dataset construction process. The paper does not describe how images and questions were selected or sourced, what categories or difficulty levels were included, or what principles guided the dataset design. These missing elements limit the benchmark’s usefulness to other researchers and reduce its potential for being extended or analyzed further.

The paper also lacks a meaningful discussion of limitations. Although the paper claims they mention limitations in the conclusion section, there is no clear section or reflection on possible biases, task design constraints, or coverage gaps in the dataset. Without such discussion, readers are left without the context needed to interpret the results or judge how generalizable the findings might be.

Another limitation is the minimal insight provided into model behavior. While Table 3 shows overall poor model performance and a single example illustrates failure on a maze task, the paper does not offer a deeper breakdown of what kinds of visual reasoning are most difficult. There is no analysis of failure patterns, nor is there a taxonomy of reasoning types present in the dataset. This makes it harder to assess the practical value of the benchmark beyond simply knowing that models perform poorly.

While a full model behavior study may be out of scope, especially for a dataset paper, some basic categorization or a few qualitative examples would help make the results more informative and actionable. Currently, the paper contains many figures just trying to highlight the difference between the current paper and similar other benchmarks, or the redundant figure one, that considering table 3, is not necessary to be included in the paper. All this space can be used for more insightful analysis or details.

**Strengths Contributions:**

The paper is clearly written and well-organized. The main contribution, which emphasizes visual reasoning rather than perception, is presented in a way that makes the motivation and novelty of the benchmark easy to understand.

The dataset itself is valuable because it represents a type of task that has been underrepresented in prior evaluation work. Instead of asking models to describe or localize elements in an image, the tasks in this benchmark require actual reasoning within the visual modality. This distinction makes the benchmark a timely and important addition, especially as interest grows in building models that can go beyond surface-level image understanding.

---

> ### Author Rebuttal · Authors · 2025-07-29
>
> Thanks for your valuable feedback. We are especially grateful for your recognition of the motivation, novelty, benchmark value, and writing quality of our paper. Meanwhile, we agree with your constructive suggestions on dataset construction, the discussion of limitations, and the importance of in-depth analysis. These insights have played a crucial role in improving the overall quality of our work. The followings are points-to-points respones to your concerns:
>
> >**Q1. For a dataset-centric paper, there is insufficient detail about the dataset construction process**
>
> A1: Thanks for this important and constructive feedback. While Section 3.1 of our paper outlines the category-wise design principles, we agree that a more systematic explanation is needed. Below, we provide additional details that will be included in the revision
>
> **Data source:**   The sources vary by category, and all problems are independently verified by at least two experts.
>
>
> 1. Math:  The data sources for each subclass in mathematics are different:
>    1. **Transformation Geometry**: Derived from high school exercises and rule-based code generation.
>    2. **Planar Geometry and Solid Geometry**: Taken from school/university exercises and math olympiads.
>    3. **Graph Theory**: From school/university exercises and competition problems.
> 2. Physics:  The physics problems are curated from a unified set of sources, ranked in a hierarchy that prioritizes problem clarity and solution reliability over quantity. Each problem was validated by Ph.D.-Level experts to ensure it necessitates drawing to solve. The sources are ranked as follows:
>    1. **Physics Olympiads**: Official competition problems from international and regional physics contests, including IPhO, APhO, EPhO and CPhO.
>    2. **University Textbooks**: Exemplary problems and exercises extracted from widely adopted undergraduate and graduate-level physics textbooks.
>    3. **Published Problem Collections**: Curated physics problem compilations from universities and academic institutions, with difficulties targeted at the final-year undergraduate level.
>    4. **Course Materials**: Tests and exercises sourced from university and high school physics courses.
> 3. Counting:  This part of data mainly comes from the Internet. We will collect some pictures about groups of animals and select the parts that need to be marked in the image to complete the counting questions (we filter out about 78% of the data); then we will make the answer.
> 4. Games:  The sources of game data are also rich and varied.
>    1. **Connect-the-dots**: This data comes from children's picture books. We selected them and filtered out those pictures that were still unclear after connecting them.
>    2. **Mazes** : This part of the data comes from children's books and online maze games. We will create some QA pairs based on the trajectory.
>    3. **Dart-and-balloon, Gold Miner, Puzzles and Ball-and-brick:** Sourced from gameplay screenshots or rule-based generation.
>
> **Principles guided the dataset design:**  The dataset design principles primarily focus on **quality** and **diversity**:
>
> 1. **Quality**: We aim for every question in the dataset to require reasoning based on modifications to the image. Since this is inherently subjective, we engage human experts to ensure the quality. For math and physics questions, we invited Ph.D.-level domain experts to perform quality control, ensuring that each problem genuinely requires visual reasoning to solve. For counting and game-related tasks, the authors themselves carefully reviewed and filtered each question to achieve the same goal. Notably, every question has been verified by at least two different experts.
> 2. **Diversity**: We strive to ensure that the dataset covers a wide range of categories, rather than being confined to a single domain such as math. Within each domain, we also include a variety of problem types, for example, the game category includes Connect-the-dots, Mazes, Dart-and-balloon, Gold Miner, Puzzles, and Ball-and-brick tasks.
>
>
> >**Q2: The paper also lacks a meaningful discussion of limitations**
>
> A2: Thanks for your suggestions. We agree that benchmark papers should include its limitations. Our paper included shortcomings of current models, but lacked a discussion of the limitations of the dataset itself. Here, we discuss the limitations of our benchmark.
>
> 1. **Modality coverage**: RBench-V currently focuses on text and image. It does not cover other modalities (e.g., audio, video), which limits its applicability to broader multi-modal tasks.
> 2. **Generalizability coverage**: Although we cover a range of fields, we still have a gap from more generalized fields such as GUI, Robotics, and we hope to establish a more comprehensive visual reasoning benchmark in a wider range of fields in the future.
> 3. **Lack of training data**: Since the data in RBench-V is manually curated rather than automatically collected, it is challenging to obtain a large-scale training set. Leveraging large-scale data through supervised fine-tuning (SFT) or reinforcement learning (RL) may offer a promising direction for addressing visual reasoning.
>
> Finally, we hope that our follow-up work can improve the above limitations and continuously improve this work.
>
> >**Q3: Another limitation is the minimal insight provided into model behavior**
>
> A3: Thank you for your constructive suggestion. We agree that analyzing model behavior provides valuable insight and enhances the practical value of our benchmark. Accordingly, we conducted detailed analyses on GPT-4o, Gemini 2.5 pro, and Qwen2.5VL-72B, covering both multi-choice and open-ended tasks, and further examined the web-based GPT-4o (o3) due to its unique visual manipulation capability.
>
>
> 1. Multi-Choice Analysis (Primarily Games)
>
> We used one-tailed binomial tests to assess whether model performance in each Game subcategory outperformed random guessing. The null hypothesis ($H_0$) posits that the model's accuracy is indistinguishable from chance, whereas the alternative hypothesis ($H_1$) is that the model's accuracy exceeds chance. A p-value below the significance level of α=0.05 was considered evidence to reject $H_0$ in favor of $H_1$. The resulting p-values are summarized in the table below:
>
> | Model           | Connect-the-dots | Mazes | Dart-and-balloon | Gold Miner | Puzzles | Ball-and-brick |
> | --------------- | ---------------- | ----- | ---------------- | ---------- | ------- | -------------- |
> | GPT-4o          | 0.761            | 0.173 | 0.936            | 0.052      | 0.489   | 0.200          |
> | Gemini  2.5 pro | 0.858            | 0.998 | 0.063            | 0.341      | 0.489   | 0.102          |
> | Qwen2.5VL-72B   | 0.761            | 0.479 | 0.063            | **0.015**  | 1.000   | 0.871          |
>
> Across the 18 tests, the p-values provided insufficient statistical evidence that models outperform random chance, as the single p-value (0.015) lower than 0.05 was not significant after a Benjamini–Hochberg correction.
>
> Conclusion: Models perform close to chance on multi-choice tasks.
>
> 2. Open-Ended Analysis (Math and Physics)
>
> Our analysis reveals a significant gap: **models consistently bypass visual reasoning by employing text-based shortcuts.** This behavior contrasts with human experts, who frequently rely on sketching to solve such problems. We identify two primary types of shortcuts:
>
> - **Exploiting Extensive Prior Knowledge.** Using memorized theorems and pretraining/web data to bypass visual reasoning. E.g., solving mechanics problems by invoking invariants without drawing.
>
>
> - **Employing Algebraic Methods.** Converting geometric/physical problems into equations. E.g., applying Kirchhoff's laws or Lagrangians instead of diagrams.
>
> This reliance on shortcuts is remarkably effective in some cases. For instance, by leveraging its strong text-based reasoning, Gemini 2.5 pro solves 43% of the math problems and 10% of the physics problems in our benchmark.
>
> 3. Visual Reasoning Patterns of OpenAI o3
>
> The web-version o3 model is of particular interest as it demonstrates the ability to interleave image manipulation within its Chain of Thought (CoT). We identified three primary visual reasoning patterns:
>
> - **Image Cropping.** The model frequently crops the input image to analyze distinct regions individually before integrating the findings. While this strategy is applied across all categories, its utility is limited. For instance, in counting problems, while the intent is to simplify the task by focusing on sub-regions, the cropping and integration process often introduces errors, such as double-counting or omissions.
>
> - **Labeling and Coordinate Embedding.** o3 typically begins by applying visual annotations, such as coordinate systems and graph-structure labels, which are then translated into textual representations for symbolic reasoning.
>
> - **Auxiliary Line Drawing.** As illustrated in Figure 3, o3 exhibits a promising capability in the *Games-mazes* task. The model recognizes the maze structure, programmatically computes the correct path, and draws this path as an auxiliary line onto the image. Finally, it answers the question by observing the augmented image. This demonstrates a promising instance of multi-modal reasoning. We believe that future work advancing the application of auxiliary line drawing within M-CoT could lead to substantial performance gains on the challenging problems curated in RBench-V.
>
> We believe the above analysis offers a clearer and more nuanced understanding of model behavior on RBench-V, and we will incorporate them into the revised version.
>
> >**Q4: A few qualitative examples would help make the results more informative and actionable**
>
> A4: Thanks for the suggestion. In the revised version, we will include additional qualitative visualizations and analyses, aligned with our responses to above concerns, to help readers better understand the model capabilities that RBench-V is designed to assess.

---

> > ### Comment · Reviewer_REov · 2025-08-03
> >
> > I thank the authors for their comprehensive reply to my comments.
> > These points address all my comments. Hence, I will update my score.
> > Please ensure that you also provide links to the sources from which you have obtained data points.
> > Furthermore, the qualitative analysis you provided here adds much more depth to the paper. Please incorporate these and potentially even more insights, as time allows, into the camera-ready version.

---

> > > ### Author Response · Authors · 2025-08-04
> > >
> > > We sincerely thank you for your positive and encouraging feedback. We are glad that our responses addressed your concerns.
> > >
> > > As suggested, we will make sure to include links to the sources of the data points in the final version. We will also incorporate the insights from our rebuttal into the final version of the paper to enhance its clarity and depth. Furthermore, we plan to carry out additional analyses. Through this process, we have realized that examining both the dataset and model behaviors offers valuable guidance for future research directions. We will include these insights in the revised version as well.
> > >
> > >
> > > Thank you again for your valuable comments and thoughtful suggestions, which have been helpful not only for improving our current work but also for guiding our future work.

---

### Official Review · Reviewer_rKFn · 2025-06-30

**Rating:** 5
**Confidence:** 4

**Summary:**

The paper introduces RBench-V, a new benchmark designed to evaluate multi-modal large language models’ abilities to perform vision-indispensable reasoning that requires generating multi-modal outputs. Unlike prior benchmarks that focus primarily on reasoning over multi-modal inputs with textual outputs, RBench-V emphasizes the model’s capability to process and generate visual content during reasoning.

**Additional Feedback:**

The open-source models evaluated in the paper can only produce text outputs. It would be helpful to clarify how multi-modal outputs are achieved in this context. Additionally, it may be valuable to consider evaluating open-source models that can generate image outputs, such as o3, BLIP-3o [1] and  DeepEyes [2], which could be more aligned with the benchmark’s focus on multi-modal output reasoning.

---
[1] BLIP3-o: A Family of Fully Open Unified Multimodal Models-Architecture, Training and Dataset, 2025.

[2] DeepEyes: Incentivizing "Thinking with Images" via Reinforcement Learning, 2025.

**Dataset Code Accessibility:**

Yes

**Dataset Code Comments:**

The dataset and code are publicly available at https://evalmodels.github.io/rbenchv. The benchmark is well-documented with clear details on tasks and evaluation, supporting reproducibility and further research use.

**Ethical Considerations:**

No, there are no or only very minor ethics concerns

**Final Justification:**

The authors have addressed my concern about various prompt designs and the performance of MLLMs with M-CoT. Overall, I incline to accept this paper.

**Limitations Weaknesses:**

1. The paper could clarify how multi-modal outputs are utilized during the reasoning process. For example, in the mazes tasks, it remains unclear whether the models employed multi-round M-CoT approaches where image outputs are fed back into the model to generate the final text answer.
2. Including more examples of successful cases of reasoning with image outputs could make the evaluation process of multi-modal reasoning outputs clearer and more convincing.
3. The paper reports comparisons with human experts, but it would be useful to clarify whether the human evaluations were conducted multiple times and averaged to ensure the stability and robustness of the results.
4. It would also be valuable to know whether prompt design was used to guide the models toward producing multi-modal outputs during evaluation.

**Strengths Contributions:**

1. The paper is well-organized and easy to follow
2. The paper innovatively highlights the importance of evaluating multimodal outputs in VLMs reasoning tasks
3. The experiments are extensive and the analysis provides some valuable insights

---

> ### Author Rebuttal · Authors · 2025-07-30
>
> Thank you for your thoughtful and encouraging feedback. We are especially grateful for your recognition for our motivation, experiments, and writing. The followings are points-to-points respones to your concerns:
>
> >**Q1: The paper could clarify how multi-modal outputs are utilized during the reasoning process. For example, in the mazes tasks, it remains unclear whether the models employed multi-round M-CoT approaches where image outputs are fed back into the model to generate the final text answer.**
>
> Thanks for raising this insightful point. We agree that multi-round Multi-modal Chain-of-Thought may be an effective method for interleaved image-text reasoning. Firstly, except for OpenAI o3, we did not observe this multi-round M-CoT behavior in other models. As for o3, we conducted a detailed analysis of its interleaved visual-textual reasoning in following.
>
> o3 model is of particular interest as it demonstrates the ability to interleave image manipulation within its M-CoT. We identified three primary visual reasoning patterns:
>
> - **Image Cropping.** The model frequently crops the input image to analyze distinct regions individually before integrating the findings. While this strategy is applied across all categories, its utility is limited. For instance, in counting problems, while the intent is to simplify the task by focusing on sub-regions, the cropping and integration process often introduces new errors, such as double-counting or omissions.
>
> - **Labeling and Coordinate Embedding.** o3 typically begins by applying visual annotations, such as coordinate systems and graph-structure labels, which are then translated into textual representations for symbolic reasoning.
>
> - **Auxiliary Line Drawing.** As illustrated in Figure 3 of our paper, o3 exhibits a promising capability in the *Games-mazes* task. The model first recognizes the maze structure, programmatically computes the correct path, and then draws this path as an auxiliary line onto the image. Finally, it answers the question by observing the augmented image. This demonstrates a promising instance of multi-modal reasoning. We believe that future work advancing the application of auxiliary line drawing within M-CoT could lead to substantial performance gains on the challenging problems curated in RBench-V.
>
> The above is our analysis of how multi-modal outputs are utilized during the reasoning process.
>
> >**Q2: Including more examples of successful cases of reasoning with image outputs could make the evaluation process of multi-modal reasoning outputs clearer and more convincing.**
>
> A2: Thank you for the constructive suggestion. We agree that including more examples of successful reasoning with image outputs would make the evaluation process of RBench-V clearer. Similar to the previous issue, we found that only o3 demonstrated correct interleaved image-text reasoning. Therefore, we will include several successful examples from o3 in the revised version.
>
> Besides, through this problem, we also analyzed the model output more deeply and discovered more text-based shortcuts.  We conducted detailed analyses on GPT-4o, Gemini 2.5 pro, and Qwen2.5VL-72B, covering both multi-choice and open-ended tasks, and further examined the web-based GPT-4o (o3) due to its unique visual manipulation capability.
>
> 1. Multi-Choice Analysis (Primarily Games)
>
> We used one-tailed binomial tests to assess whether model accuracy in each Game subcategory exceeds random chance. The null hypothesis, $H_0$, posits that the model's performance is indistinguishable from chance. A p-value below the significance level (α=0.05) allows for the rejection of $H_0$, suggesting that the model's performance is significantly better than random guessing. The resulting p-values are summarized in the table below:
>
> | Model           | Connect-the-dots | Mazes | Dart-and-balloon | Gold Miner | Puzzles | Ball-and-brick |
> | --------------- | ---------------- | ----- | ---------------- | ---------- | ------- | -------------- |
> | GPT-4o          | 0.761            | 0.173 | 0.936            | 0.052      | 0.489   | 0.200          |
> | Gemini  2.5 pro | 0.858            | 0.998 | 0.063            | 0.341      | 0.489   | 0.102          |
> | Qwen2.5VL-72B   | 0.761            | 0.479 | 0.063            | **0.015**  | 1.000   | 0.871          |
>
> Across the 18 tests, the vast majority of p-values are substantially greater than 0.05. According to the Benjamini–Hochberg correction, the single low p-value (0.015) is likely a false positive. This indicates insufficient statistical evidence to conclude that the models perform better than random chance.
>
> Conclusion: Models perform close to chance on multi-choice tasks.
>
> 2. Open-Ended Analysis (Math and Physics)
>
> Our analysis reveals a significant gap: **instead of leveraging multi-modal outputs for visual reasoning, models consistently resort to text-based shortcuts.** Unlike human experts who often rely on sketching to solve math and physics problems, we observe that models predominantly resort to text-based shortcuts, circumventing the need for image manipulation. We identify two primary types of shortcuts:
>
> - **Exploiting Extensive Prior Knowledge.** Using memorized theorems and pretraining/web data to bypass visual reasoning. E.g., solving mechanics problems by invoking invariants without drawing.
>
>
> - **Employing Algebraic Methods.** Converting geometric/physical problems into equations. E.g., applying Kirchhoff's laws or Lagrangians instead of diagrams.
>
> This reliance on shortcuts is remarkably effective in some cases. For instance, by leveraging its strong text-based reasoning, Gemini 2.5 pro solves 43% of the math problems and 10% of the physics problems in our benchmark.
>
> Thank you once again for your insightful comment. We believe it has greatly deepened our understanding of both RBench-V and the models involved.
>
>
>
> >**Q3: The paper reports comparisons with human experts, but it would be useful to clarify whether the human evaluations were conducted multiple times and averaged to ensure the stability and robustness of the results.**
>
> A3: Thanks for pointing it out.  In this paper, Each question was answered by a single annotator within its domain (e.g., physics questions were completed by physics students). We agree that performing multiple evaluations per question would ensure stability and robustness. In rebuttal stage, due to time constraints, we conducted two additional rounds of human evaluation on the counting subset only. The resulting accuracies were 82.6% and 83.6%, respectively. Combined with our initial result of 81.0%, the average accuracy across the three runs is 82.4%.
>
> This indicates that single-pass human evaluation does introduce some bias. We plan to conduct further human assessments to ensure that each question in RBench-V is evaluated by three independent annotators.
>
>
>
> >**Q4: It would also be valuable to know whether prompt design was used to guide the models toward producing multi-modal outputs during evaluation.**
>
> A4: Thank you for this insightful question. To address it, we designed four different prompts as outlined below and conducted experiments with GPT-4o, Qwen2.5-VL-7B, LLaVA-OV-7B, and LLaVA-OV-72B.
>
> Prompt A (in original paper): Solve the following question.
>
> Prompt B: Solve the following question according to the given picture step-by-step.
> Prompt C: Solve the following question according to the given picture. Give your final result directly without any thinking process.
> Prompt D: Solve the following question according to the given picture step-by-step. Note that in order to solve this problem, you may need to think with image.
>
> |          | GPT-4o   | Qwen2.5-VL-7B | LLaVA-OV-7B | LLaVA-OV-72B |
> | -------- | -------- | ------------- | ----------- | ------------ |
> | Prompt A | 14.1     | 8.3           | 8.5         | 9.0          |
> | Prompt B | 13.3     | **9.3**       | 8.6         | **10.0**     |
> | Prompt C | 12.5     | 8.8           | 8.2         | 9.6          |
> | Prompt D | **14.3** | 7.6           | **8.8**     | 9.2          |
>
> From the table above, we can see that the prompt has little impact on the results. Prompt engineering may not be an effective way to solve visual reasoning problems.
>
>
>
> >**Q5: The open-source models evaluated in the paper can only produce text outputs. It would be helpful to clarify how multi-modal outputs are achieved in this context.  Additionally, it may be valuable to consider evaluating open-source models that can generate image outputs, such as o3, BLIP-3o and DeepEyes, which could be more aligned with the benchmark’s focus on multi-modal output reasoning.**
>
> A5: Thanks for this helpful suggestion.We fully agree that evaluating models that can natively generate image outputs is crucial for a benchmark like RBench-V, which emphasizes multi-modal output reasoning. Therefore, we evaluated both BLIP3-o and DeepEyes, and the results are presented in the table below. We will **definitely** integrate above BLIP3-o and DeepEyes into the experiments and related work sections of the revised version to make the paper more complete and comprehensive.
>
> | Model       | Overall | Math  | Physics | Counting | Game  |
> | ----------- | ------- | ----- | ------- | -------- | ----- |
> | BLIP3o-8B   | 8.7%    | 12.5% | 1.9%    | 4.1%     | 13.5% |
> | DeepEyes-7B | 8.6%    | 15.3% | 2.5%    | 2.6%     | 12.0% |
>
> The above experimental results indicate that even native multimodal models, which can generate multi-modal outputs, still struggle to effectively address the challenges of visual reasoning. In future work, we will continue to explore effective strategies for solving interleaved image-text reasoning.

---

> > ### Comment · Reviewer_rKFn · 2025-08-01
> >
> > I appreciate the authors' response. Most of my concerns have been resolved. However, regarding Q4, it seems that the improvement brought by the prompt D method, which guides image-based thinking, is quite limited. Is this because the model cannot output images to support multi-turn M-CoT? Overall, I will raise my score to 5.

---

> > > ### Author Response · Authors · 2025-08-01
> > > **Response to reviewer rKFn**
> > >
> > > Thanks for your timely and positive response. We’re pleased to see that the rebuttal has addressed most of your concerns and that you are willing to raise your score to 5, which is incredibly encouraging for us.
> > >
> > > >**For your question, regarding Q4, it seems that the improvement brought by the prompt D method, which guides image-based thinking, is quite limited. Is this because the model cannot output images to support multi-turn M-CoT?**
> > >
> > > We conducted additional experiments comparing **Prompt A** and **Prompt D** on **BLIP3o-8B** and **DeepEyes-7B**, as these models are capable of generating image outputs. The experimental results are as follows:
> > >
> > > |          | BLIP3o-8B | DeepEyes-7B |
> > > | -------- | --------- | ----------- |
> > > | Prompt A | 8.7%      | 8.6%        |
> > > | Prompt D | 8.5%      | 7.1%        |
> > >
> > > The experimental results indicate that even models capable of generating images did not achieve significant improvements under Prompt D.
> > >
> > > Our analysis suggests that this may be due to the high level of precision required for image generation in RBench-V. For example, models are expected to accurately trace a path through a maze or precisely connect two points without affecting other parts of the image. This goes beyond coarse or approximate image generation and places greater demands on the model’s ability to produce fine-grained, controlled outputs, a capability that current models still struggle with.
> > >
> > > A simple and easily reproducible experiment is to ask GPT-4o to connect two points, such as E and F, in a geometric diagram. GPT-4o fails to accurately complete this task without affecting other parts of the diagram.
> > >
> > > In summary, we agree that visual prompting is a promising direction for addressing M-CoT, and we will continue to explore it further. Additionally, we believe that fine-grained and accurate image generation strategies will also play a crucial role in enabling effective M-CoT.
> > >
> > > Overall, we sincerely appreciate your timely and positive response, and we hope the above reply can address your concern.

---

> > > > ### Comment · Reviewer_rKFn · 2025-08-01
> > > >
> > > > Thank you for the explanation. I have updated my rating.

---

> > > > > ### Author Response · Authors · 2025-08-01
> > > > >
> > > > > Thanks for improving your rating.

---

### Official Review · Reviewer_MM7V · 2025-06-30

**Rating:** 4
**Confidence:** 3

**Summary:**

This paper introduces RBench-V, a novel benchmark designed to evaluate visual reasoning models with multi-modal output capabilities. The benchmark includes many carefully curated questions across four domains (math, physics, counting, and games) that require models to generate multi-modal outputs, such as drawings and image modifications, during problem-solving. The results show that even the best-performing model achieves only 25.8% accuracy, far below the human score of 82.3%, highlighting the significant gap in current models' ability to handle multi-modal visual reasoning tasks.

**Dataset Code Accessibility:**

Yes

**Ethical Considerations:**

No, there are no or only very minor ethics concerns

**Final Justification:**

I maintain the score

**Limitations Weaknesses:**

1. In Figure 3, the authors state, "The red lines shown in the figure are not part of the original questions." Therefore, I believe these elements should not be included under the "question" section but instead placed under the "answer" section to avoid potential misunderstandings.

2. Could the authors provide visualizations of the complete multi-modal reasoning process for some examples from the proposed benchmark? This would help clarify how the models approach and solve the tasks.

**Strengths Contributions:**

+ The paper provides a clear motivation for RBench-V, emphasizing its focus on multi-modal outputs, which is underexplored in existing benchmarks.
+The benchmark covers a wide range of tasks (math, physics, counting, and games), challenging models to engage in complex visual reasoning.

---

> ### Author Rebuttal · Authors · 2025-07-26
>
> We sincerely thank the reviewers for their selfless efforts and constructive feedback, which have greatly helped us improve the quality of this work. We are particularly grateful for your recognition of the core motivation and novelty of our focus on multi-modal output reasoning for MLLMs, an area we believe is critical yet underexplored.. Below, we provide point-by-point responses for your concerns:
>
>
> >**Q1 : In Figure 3, the authors state, "The red lines shown in the figure are not part of the original questions." Therefore, I believe these elements should not be included under the "question" section but instead placed under the "answer" section to avoid potential misunderstandings.**
>
> A1 : Thanks for your careful reading and for pointing this out. In Figure 3, the red lines (e.g., auxiliary lines) were not part of the original question input, which may indeed cause confusion for readers. To address this, we will revise the figure in the revision by relocating the auxiliary lines to the answer column to avoid misunderstandings. We believe this adjustment will improve clarity and reduce potential misunderstandings. Due to limitations of the neurips rebuttal stage, we are unable to provide visualization results, but we are sure that the above changes will appear in the next version.
>
>
> >**Q2:  Could the authors provide visualizations of the complete multi-modal reasoning process for some examples from the proposed benchmark? This would help clarify how the models approach and solve the tasks.**
>
>
> A2 : Thank you for the constructive suggestion. We strongly agree that providing some visualizations of the complete multi-modal reasoning process would greatly improve the quality of this work. In revision, we will include visualized multi-modal reasoning paths for different types of questions, both in the main text and the appendix. We believe this addition will help readers gain a clearer and deeper understanding.Thank you again for this very helpful suggestion. Due to space and format constraints during the NeurIPS rebuttal stage, we are unable to include these visualizations here. However, we are **definitely** sure it will be included in the revised version.
>
> Thank you once again for your positive recognition and your insightful and valuable suggestions, which have greatly improved the quality of our work. All the corresponding revisions will be fully reflected in the revised version.

---

### Official Review · Reviewer_1qaj · 2025-07-03

**Rating:** 5
**Confidence:** 3

**Summary:**

This paper proposes RBench-V, a benchmark specifically designed to evaluate multi-modal output reasoning, which is an underexplored but increasingly important capability for LMMs. The benchmark includes 803 questions from math, physics, counting, and games that require generating visual outputs (e.g., drawing auxiliary lines, tracing paths in mazes). A wide range of models, including GPT-4o, Claude, Gemini, Qwen, and open-source baselines, are evaluated. The authors further analyze performance differences across task types, modalities, model sizes, and reasoning styles.

**Dataset Code Accessibility:**

Yes

**Ethical Considerations:**

No, there are no or only very minor ethics concerns

**Final Justification:**

All of my concerns have been addressed clearly and effectively. I will be maintaining the positive rating as a result.

**Limitations Weaknesses:**

- Though practical, this benchmark uses LLM-as-a-judge. It may introduces potential bias (depends on GPT-4o as a judge) and lacks formal alignment analysis with human raters.
- All evaluations are zero-shot; it would be useful to test if training on visual reasoning with supervised or RL-based objectives improves performance.
- As outputs are visual, the evaluation is less standardized than traditional benchmarks, which may affect the reproducibility of this benchmark.

**Strengths Contributions:**

- The authors rightly identify the gap that current benchmarks mostly assess perception and textual reasoning, while ignoring visual output reasoning, which is key in many real-world tasks. This is arguably the first benchmark that explicitly requires visual CoT-style reasoning and multi-modal output (e.g., drawing, connecting, manipulating figures).
- The proposed benchmark covers a wide spectrum of tasks that demand genuine visual manipulation (geometry, optics, circuits, puzzles).
- The paper includes open- and closed-source models, human experts, and includes "w/o math" subset analysis to avoid model shortcutting. The evaluation discusses how models tend to use algebraic shortcuts instead of visual reasoning, and includes strong visual illustrations and failure analysis.

---

> ### Author Rebuttal · Authors · 2025-07-30
>
> We sincerely thank the reviewers for their selfless efforts and constructive feedback, which have greatly helped us improve the quality of this work. We are especially grateful for the recognition of visual chain-of-thought (CoT) reasoning as a critical yet underexplored capability for MLLMs. Below, we provide point-by-point responses for your concerns:
>
> >**Q1: Though practical, this benchmark uses LLM-as-a-judge. It may introduces potential bias (depends on GPT-4o as a judge) and lacks formal alignment analysis with human raters.**
>
> A1: Thanks for raising this important concern. We fully agree that using LLM-as-a-judge, while practical and widely adopted, may introduce evaluation bias. To address this issue, we conducted a human evaluation experiment and compared the results with GPT-4o's judgments. We designed experiments on RBench-V involving GPT-4o, Gemini 2.5 pro, and Qwen2.5VL-72B. The results are showing in following table.
>
> | Model                         | Math | Physics | Counting | Game | Overall |
> | ----------------------------- | ---- | ------- | -------- | ---- | ------- |
> | GPT-4o  (GPT-4o Judge)        | 24.4 | 3.2     | 13.3     | 14.2 | 14.1    |
> | GPT-4o (Human Judge)          | 24.4 | 3.2     | 13.3     | 13.8 | 13.9    |
> | Gemini 2.5 pro (GPT-4o Judge) | 42.6 | 9.6     | 19.0     | 12.7 | 20.2    |
> | Gemini 2.5 pro (Human Judge)  | 42.6 | 11.5    | 19.0     | 12.7 | 20.5    |
> | Qwen2.5VL-72B (GPT-4o Judge)  | 15.3 | 3.8     | 6.2      | 14.5 | 10.6    |
> | Qwen2.5VL-72B (Human Judge)   | 15.3 | 3.8     | 6.0      | 14.2 | 10.3    |
>
> As shown in the table, GPT-4o’s judgments align with human evaluations in most cases. This is largely because the answers to most questions are either a number or a multiple-choice option. An exception occurs with Gemini 2.5 pro’s accuracy on physics problems: GPT-4o mistakenly judged three correct answers as incorrect. We found this was due to some physics answers involving relatively complex formulas, where differences in expression order or simplification led GPT-4o to make incorrect evaluations.
>
>
>
> >**Q2: All evaluations are zero-shot; it would be useful to test if training on visual reasoning with supervised or RL-based objectives improves performance.**
>
> A2: Thank you for the thoughtful suggestion. We agree that exploring whether supervised fine-tuning (SFT)  or reinforcement learning (RL) on visual reasoning tasks improves model performance on RBench-V is a promising and important direction.
>
> Since the data collected in RBench-V does not contain intermediate reasoning steps, we choose to train the model using the GRPO reinforcement learning algorithm [1]. All experiments are conducted using **Qwen2.5VL-7B** as the base model with 50 epochs.
>
> We design two experimental settings based on different data partitions:
>
> **Setting 1**: We split the entire RBench-V dataset into 80% for training and 20% for testing. We observe that although applying GRPO significantly improves accuracy on the training set (overfitting), this performance does **not generalize** to the test set. The detailed results are shown in the table below:
>
> |              | Before training | After training |
> | ------------ | --------------- | -------------- |
> | Training Set | 7.7%            | 15.7%          |
> | Test Set     | 7.8%            | 8.4%           |
>
> **Setting 2**: We focus on the **Counting** subset of RBench-V, similarly splitting it into 80% for training and 20% for testing. Again, we find that while GRPO improves performance considerably on the training set, this gain **fails to transfer** to the test set. Results are summarized as follows:
>
> |              | Before training | After training |
> | ------------ | --------------- | -------------- |
> | Training Set | 3.8%            | 8.8%           |
> | Test Set     | 2.9%            | 2.9%           |
>
>
>
> These findings suggest that **visual reasoning may not be easily acquired through simple reinforcement learning techniques**. We will continue exploring alternative methods to enhance the model’s capability in visual reasoning in future work.
>
> [1]. Guo D, Yang D, Zhang H, et al. Deepseek-r1: Incentivizing reasoning capability in llms via reinforcement learning[J]. arXiv preprint arXiv:2501.12948, 2025.
>
> >**Q3: As outputs are visual, the evaluation is less standardized than traditional benchmarks, which may affect the reproducibility of this benchmark.**
>
> A3: We sincerely apologize for confusion caused by unclear expressions in our paper, which may have led to a misunderstanding of our evaluation process.
>
> RBench-V is primarily designed to assess reasoning through multimodal outputs, where we encourage models to generate interleaved visual and textual reasoning steps during problem solving. However, due to current evaluation constraints, all final answers in RBench-V are in **text format**. As a result, the evaluation of final outputs follows the same procedure as prior benchmarks, which is reproducible and consistent, and we have integrated our evaluation pipeline into standard toolkits such as vlmevalkit and lmms-eval to facilitate reproducibility.
>
> Additionally, we have conducted further experiments during the rebuttal phase and confirmed that the evaluation results are stable and reproducible across multiple runs.
>
> Once again, we apologize for the confusion and will revise the paper to clarify this point more explicitly in the revised version.

---

> > ### Comment · Reviewer_1qaj · 2025-08-06
> >
> > Thanks for the clear and detailed response. You've addressed all of my concerns clearly and effectively. I appreciate the effort you've put into this, and I will be maintaining the positive rating as a result.

---

> > > ### Author Response · Authors · 2025-08-07
> > >
> > > Thanks for keeping your highly positive score.

---

### Official Review · Reviewer_rrEA · 2025-07-20

**Rating:** 5
**Confidence:** 4

**Summary:**

This paper introduces RBench-V, a benchmark specifically designed to evaluate the visual reasoning abilities of multi-modal models, with a strong focus on tasks that require generating multi-modal outputs. The authors hand-picked 803 diverse questions spanning math, physics, counting, and games, each demanding models to manipulate images as part of the reasoning process. Their experiments show that current state-of-the-art models, including GPT-4o, Gemini, and o3, perform far below human levels, especially in tasks that need genuine visual reasoning.

**Dataset Code Accessibility:**

Yes

**Ethical Considerations:**

No, there are no or only very minor ethics concerns

**Final Justification:**

Most concerns have been addressed. Therefore, I recommend to accept this work.

**Limitations Weaknesses:**

- The models sometimes “solve” visual questions using text-based shortcuts, so high scores in some categories may not reflect real visual reasoning.
- This benchmark is still limited to text and image modalities and does not address tasks involving other types of outputs, such as audio or video.

**Strengths Contributions:**

This paper successfully identifies and addresses a critical yet underexplored gap in AI evaluation, shifting the focus from multimodal input understanding to the generation and manipulation of multimodal outputs. The carefully curated question set in RBench-V genuinely requires models to perform visual-spatial operations, effectively preventing models from “shortcutting” the problems through simple text or pattern recognition. Be specific:
- The benchmark is carefully curated to test real visual reasoning, not just multi-modal input understanding, forcing models to generate and manipulate images as part of the answer.
- The evaluation covers a wide variety of problem types and includes comparisons across multiple leading models, as well as human performance, providing a clear view of where current AI stands.
- By including detailed breakdowns by question type, the paper reveals not just overall model performance but also where the biggest gaps are.

---

> ### Author Rebuttal · Authors · 2025-07-29
>
> Thank you for your thoughtful and encouraging feedback. We are especially grateful for your recognition of the core motivation, shifting the focus from multimodal input understanding to the generation and manipulation of multimodal outputs (a.k.a vision-indispensable reasoning), behind this work. The followings are points-to-points respones to your concerns:
>
> >**Q1: The models sometimes “solve” visual questions using text-based shortcuts, so high scores in some categories may not reflect real visual reasoning.**
>
>  A1 : Thank you for your constructive suggestion. We agree that text-based shortcuts can lead to some high scores. Therefore, we conducted a systematic analysis of this phenomenon and uncovered several interesting findings. Accordingly, we conducted detailed analyses on o3, GPT-4o, Gemini 2.5 pro, and Qwen2.5VL-72B, covering both multi-choice and open-ended tasks.
>
> 1. Multi-Choice Analysis (Primarily Games)
>
>     We used one-tailed binomial tests to assess whether model performance in each Game subcategory outperformed random guessing. The null hypothesis ($H_0$) posits that the model's accuracy is indistinguishable from chance, whereas the alternative hypothesis ($H_1$) is that the model's accuracy exceeds chance. A p-value below the significance level of α=0.05 was considered evidence to reject $H_0$ in favor of $H_1$. The resulting p-values are summarized in the table below:
>
>     | Model           | Connect-the-dots | Mazes | Dart-and-balloon | Gold Miner | Puzzles | Ball-and-brick |
>     | --------------- | ---------------- | ----- | ---------------- | ---------- | ------- | -------------- |
>     | GPT-4o          | 0.761            | 0.173 | 0.936            | 0.052      | 0.489   | 0.200          |
>     | Gemini  2.5 pro | 0.858            | 0.998 | 0.063            | 0.341      | 0.489   | 0.102          |
>     | Qwen2.5VL-72B   | 0.761            | 0.479 | 0.063            | **0.015**  | 1.000   | 0.871          |
>
>     Across the 18 tests, the vast majority of p-values are substantially greater than 0.05. According to the Benjamini–Hochberg correction, the single low p-value (0.015) is likely a false positive. This indicates insufficient statistical evidence to conclude that the models perform better than random chance.
>
>     Conclusion: Models perform close to chance on multi-choice tasks.
>
> 2. Open-Ended Analysis (Math and Physics)
>
>     Our analysis reveals a significant gap: **models consistently bypass visual reasoning by employing text-based shortcuts.** This behavior contrasts with human experts, who frequently rely on sketching to solve such problems. We identify two primary types of shortcuts:
>
>     - **Exploiting Extensive Prior Knowledge.** Using memorized theorems and pretraining/web data to bypass visual reasoning. E.g., solving mechanics problems by invoking invariants without drawing.
>
>
>     - **Employing Algebraic Methods.** Converting geometric/physical problems into equations. E.g., applying Kirchhoff's laws or Lagrangians instead of diagrams.
>
>     This reliance on shortcuts is remarkably effective in some cases. For instance, by leveraging its strong text-based reasoning, Gemini 2.5 pro solves 43% of the math problems and 10% of the physics problems in our benchmark.
>
> 3. Visual Reasoning Patterns of OpenAI o3
>
>     The web-version o3 model is of particular interest as it demonstrates the ability to interleave image manipulation within its Chain of Thought (CoT). We identified three primary visual reasoning patterns:
>
>     - **Image Cropping.** The model frequently crops the input image to analyze distinct regions individually before integrating the findings. While this strategy is applied across all categories, its utility is limited. For instance, in counting problems, while the intent is to simplify the task by focusing on sub-regions, the cropping and integration process often introduces new errors, such as double-counting or omissions.
>
>     - **Labeling and Coordinate Embedding.** o3 typically begins by applying visual annotations, such as coordinate systems and graph-structure labels, which are then translated into textual representations for symbolic reasoning.
>
>     - **Auxiliary Line Drawing.** As illustrated in Figure 3 of our paper, o3 exhibits a promising capability in the *Games-mazes* task. The model first recognizes the maze structure, programmatically computes the correct path, and then draws this path as an auxiliary line onto the image. Finally, it answers the question by observing the augmented image. This demonstrates a promising instance of multi-modal reasoning. We believe that future work advancing the application of auxiliary line drawing within M-CoT could lead to substantial performance gains on the challenging problems curated in RBench-V.
>
>     We believe the above analysis provides more understanding about text-based shortcut and models' visual reasoning., and we will incorporate them into the revised version.
>
> >**Q2: This benchmark is still limited to text and image modalities and does not address tasks involving other types of outputs, such as audio or video.**
>
> A2: Thank you for pointing this out. We agree that multi-modal outputs beyond images and text, such as audio or video, represent important future directions for comprehensive evaluation of foundation models.
>
> - In future work, we will try to explore more forms of output-based reasoning and continuously expand the benchmark, which may be named as RBench-o for full modal reasoning. In fact, we are also considering how to design a reasoning paradigm based on video outputs during the process of completing this paper.
> - Moreover, image-based output reasoning represents the most prevalent and technically mature form of non-text generation currently supported by advanced foundation models (e.g., GPT-4o, Gemini 2.5 pro). For this reason, we focus our benchmark on image-based reasoning tasks as a first step toward evaluating multimodal output capabilities. It also reflects the “primary” aspect highlighted in the paper’s title, marking our initial effort to expose current limitations in visual reasoning with multimodal outputs.

---

> > ### Comment · Reviewer_rrEA · 2025-08-06
> >
> > Thanks for detailed response. I will maintain the positive score.

---

> > ### Author Response · Authors · 2025-08-06
> >
> > Thanks for keeping your highly positive score.

---

### Note · Authors · 2025-08-12

We sincerely thank all reviewers, AC, SAC and PC for their efforts in improving our work. We are also pleased to see the active engagement from everyone throughout the review and rebuttal stages. Notably, we are encouraged that our core motivation, shifting the focus from multimodal input understanding to interleaved visual-textual reasoning, has been well appreciated by all reviewers.

During the rebuttal, we provided substantial clarifications and new analyses addressing key concerns and engaged in active discussions with reviewers. These included clarifying model behavior and text-based shortcuts, addressing evaluation bias, expanding dataset sources and details, discussing dataset limitations, improving qualitative insights, and exploring training strategies. Overall, the rebuttal process strengthened both the depth of the work, the technical integrity, and the clarity of presentation, and we are glad to see that all reviewers’ concerns were resolved.


In summary, we are grateful for the reviewers’ constructive suggestions, which we will incorporate into the revised version, adding richer dataset documentation, more qualitative examples, explicit limitations, improved clarity in figures, etc. We also look forward to exploring novel training paradigms that genuinely enhance interleaved visual-textual reasoning.

Moreover, we believe RBench-V fills a critical gap in benchmarking multimodal models by centering evaluation on interleaved visual-textual reasoning. We look forward to RBench-V inspiring new research directions in the community.


Finally, we once again express our sincere gratitude to the reviewers and chairs for their important contributions to improving the quality of our work.

Sincerely,

Authors

---

### Decision · Program_Chairs · 2025-09-18

**Decision:**

Accept (poster)

**Comment:**

I think this is a very nice paper that contributes a valuable benchmark to the community for focusing on multimodal reasoning. The authors put a good amount of effort into understanding where shortcuts may circumvent the need for true multimodal reasoning (which is always important for these kinds of benchmarks), which itself is an interesting contribution -- that some of these problems can be solved in other unintended ways. I'd be curious whether the authors have suggestions on how to further improve robustness to shortcuts -- it would be unfortunate if there was a class of problems that is inherently prone to such shortcuts. Maybe it's important to prompt against and/or verify the kinds of shortcuts: "don't use knowledge of theorems xyz..."

Regarding the point that line auxiliary line drawing turned out to be a useful strategy in some cases, I think it would be good to cite some previous work on visual inductive scratchpads: https://openreview.net/forum?id=wdmI6A9d2w

I'm also looking forward to seeing a visualization of the reasoning process for some examples (as mentioned by a reviewer and implicitly by others).

===== FINAL UPDATE FROM DB Track PCs ====

The final decision for this paper has been taken by the program chairs after consultation with the SACs. All Senior Area Chairs have ranked papers according to the feedback from the AC during the review process. We decided to leave the original meta-review to reflect the opinion of the AC in light of the initial discussions with reviewers and SAC.